# Reporting and methodological quality of systematic reviews and meta-analysis with protocols in Diabetes Mellitus Type II: A systematic review

**Daniel Christopher Rainkie** ⓘ *, **Zeinab Salman Abedini, Nada Nabil Abdelkader**

Qatar University Health Cluster, College of Pharmacy, Qatar University, Doha, Qatar

* drainkie@qu.edu.qa

**Data Availability Statement:** All relevant data are within the manuscript and its Supporting information files.

**Funding:** This project is funded by Qatar University Undergraduate Student Grant number QUST-1-

## Abstract

### Background

Systematic reviews with or without meta-analyses (SR/MAs) are strongly encouraged to work from a protocol to facilitate high quality, transparent methodology. The completeness of reporting of a protocol (PRISMA-P) and manuscript (PRISMA) is essential to the quality appraisal (AMSTAR-2) and appropriate use of SR/MAs in making treatment decisions.

### Objectives

The objectives of this study were to describe the completeness of reporting and quality of SR/MAs, assess the correlations between PRISMA-P, PRISMA, and AMSTAR-2, and to identify reporting characteristics between similar items of PRISMA-P and PRISMA.

### Methods

We performed a systematic review of Type 2 Diabetes Mellitus SR/MAs of hypoglycemic agents with publicly available protocols. Cochrane reviews, guidelines, and specific types of MA were excluded. Two reviewers independently, (i) searched PubMed and Embase between 1/1/2015 to 20/3/2019; (ii) identified protocols of included studies by searching the manuscript bibliography, supplementary material, PROSPERO, and Google; (iii) completed PRISMA-P, PRISMA, and AMSTAR-2 tools. Data analysis included descriptive statistics, Pearson correlation, and multivariable linear regression.

### Results

Of 357 relevant SR/MAs, 51 had available protocols and were included. The average score for PRISMA-P was 15.8±3.3 (66%; maximum 24) and 25.2±1.1 (93%; maximum 27) for PRISMA. The quality of SR/MAs assessed using the AMSTAR-2 tool identified an overall poor quality (63% critically low, 18% low, 8% moderate, 12% high). The correlation between the PRISMA-P and PRISMA was not significant (r = 0.264; p = 0.06). Correlation was significant between PRISMA-P and AMSTAR-2 (r = 0.333; p = 0.02) and PRISMA and AMSTAR-

CPH2020-17. The funder had no role in study design, data collection and analysis, decision to publish, or preparation of the manuscript.

**Competing interests:** The authors have declared that no competing interests exist.

2 (r = 0.555; p<0.01). Discrepancies in reporting were common between similar PRISMA-P and PRISMA items.

## Conclusion

Adherence to protocol reporting guidance was poor while manuscript reporting was comprehensive. Protocol completeness is not associated with a completely reported manuscript. Independently, PRISMA-P and PRISMA scores were weakly associated with higher quality assessments but insufficient as a surrogate for quality. Critical areas for quality improvement include protocol description, investigating causes of heterogeneity, and the impact of risk of bias on the evidence synthesis.

## Introduction

Systematic reviews are considered the highest form of evidence for all types of clinical questions to inform evidence-based practice [1]. When it is possible and relevant, various meta-analytic methods can be used to aggregate the results of included studies into a single point estimate and 95% confidence interval to make a conclusion of benefit, no difference, or harm. Ultimately, clinicians use the information from systematic reviews with or without meta-analyses (SR/MAs) to inform their treatment decisions. Making treatment decisions based on limited or unreliable information puts patients at risk of harm. Therefore, as part of the evidence-based process, clinicians should critically review their SR of interest to determine the accuracy and reliability of its conclusions [2].

Several checklists and tools have been developed to aid clinicians better understand the internal and external validity of their article of interest and aid authors in producing valid, reproducible results. The 2017 "A MeaSurement Tool to Assess systematic Reviews" (AMSTAR-2) tool, an update to the AMSTAR tool originally published in 2009, provides readers a systematic process to evaluate the quality of SR/MAs that includes randomized controlled trials or non-randomized studies of interventions [3]. Without adequate reporting within a published manuscript, readers of SR/MAs are unable to make accurate quality judgements. The Preferred Reporting Items for Systematic Reviews and Meta-analyses (PRISMA) statement, formerly QUality Of Reporting of Meta-analyses (QUORUM), provides instructions to authors to transparently detail their methodology and approach to coalescing the available data [4, 5]. The PRISMA checklist is required in many journals when submitting a SR/MA for publication [6]. A cross sectional study of therapeutic non-Cochrane SR/MAs in 2016 found that approximately 2 of every 3 published SR/MAs did not use a protocol or did not use PRISMA as a reporting guideline [7].

It is recommended that a pre-specified protocol should be used as the foundation for completing a SR/MA for transparent science and to identify the risk of selection and reporting bias. In 2015, the PRISMA group published several extensions to aid authors in the transparent reporting of their manuscripts. One of these extension checklists is the PRISMA-protocol (PRISMA-P) which is to be used to enhance the completeness of information presenting in a protocol of SR/MAs [8, 9]. Pre-registered systematic reviews have been associated with higher revised-AMSTAR scores compared to nonregistered reviews [10]. Although the availability of protocols for SR/MAs are rare but the presence of a protocol is associated with higher quality studies, it remains unknown how well protocols are reported in the literature and if more complete protocols associated with higher quality studies as assessed by newer and more comprehensive quality tools.

The primary outcomes of this study were to 1) assess the completeness of reporting of available protocols according to the PRISMA-P checklist, 2) the completeness of reporting of SR/MAs according to the PRISMA checklist, and 3) the quality of the SR/MAs according to the AMSTAR-2 tool. The secondary outcomes for this study were to 1) determine the correlations between PRISMA-P, PRISMA, and the overall SR/MA quality conclusions and 2) describe the discrepancies in reporting between PRISMA-P and PRISMA.

## Methods

We conducted and reported this systematic review in accordance with the PRISMA statement (S1 Checklist) [5]. A pre-specified protocol was submitted for publishing and can be found in the supplementary material in accordance with the PRISMA-P statement (S1 File) [11].

### Study population

The focus of this review was on SR/MAs of pharmacological interventions used in the management of Type 2 Diabetes Mellitus (T2DM). We chose this study population due to the widespread availability of SR/MAs assessing the plethora of medications and classes of medications on the market.

### Search strategy

A systematic search of MEDLINE (using PubMed) and Embase databases for all systematic reviews or meta-analyses investigating pharmacological interventions used in the management of T2DM for any outcome.

The MeSH terms used for PubMed database search were: ("Diabetes Mellitus, Type 2"[Mesh] AND "Hypoglycemic Agents"[Mesh] AND ((Meta-Analysis[ptyp] OR systematic [sb]) AND ("2015/01/01"[PDAT]: "2019/03/20"[PDAT]) AND "humans"[MeSH Terms] AND English[lang])).

In Embase, the Emtree terms and limits used were: ('non insulin dependent diabetes mellitus'/exp/mj AND 'antidiabetic agent'/exp/mj AND ([systematic review]/lim OR [meta analysis]/lim) AND [2015–2019]/py).

The filters applied to the searches in both databases were human only studies; and studies published in English. The language restriction of English only was applied as no translation services were available and English systematic reviews comprise the majority of the available literature. We limited our search to include only articles published between 1st January 2015 and 20th March 2019 as the PRISMA-P checklist was published in 2015.

### Eligibility criteria

The pre-specified eligibility criteria comprised of either SR/MAs with accessible protocols. We identified protocols through a search of the manuscript; bibliography; journal website for supplementary materials or appendices; the PROSPERO database; and a Google search. We did not include SR/MAs which stated that a protocol was available but the authors must be contacted. The rationale for this is that it would be rare for clinicians to access this information to completely appraise the article before interpreting and applying the results.

Cochrane reviews were excluded from our search since a peer-reviewed protocol is mandatory to publish. Other types of studies excluded from the search included published guidelines, select article meta-analyses (i.e. where no systematic review was conducted), network meta-analyses, and individual patient meta-analyses because the reporting and quality indicators specific to each of these designs are not addressed in the PRISMA-P, PRISMA or AMSTAR-2 tools.

## Study selection

The systematic search of the databases and screening of articles by title and abstract were completed independently by all three reviewers NA, ZA, and DR. After removing duplicates, if two or more reviewers identified an article for inclusion on screening, the full text was retrieved. If only a single reviewer identified an article for inclusion on screening, a final decision to retrieve the full text was based on a discussion between all the reviewers. Full texts of the screened articles were collected, and the complete pre-specified inclusion (e.g. protocol availability) and exclusion criteria were applied. We initially intended to use a random sample of 100 eligible articles to be included in our analysis; however, given the limited availability of SR/MAs with available published protocols, we included all relevant articles.

## Data management and extraction

All authors read and studied the explanation and elaboration documents associated with each of the PRISMA-P, PRISMA, and AMSTAR-2 checklists/tools in order to ensure the original intents were kept [3–5, 8, 9, 11–14]. The PRISMA-P checklist has 17-items for assessing protocols, while PRISMA checklist has 27-items for assessing the manuscript itself. The AMSTAR-2 checklist has 16-items, including 7 critical domains, used to assessing the quality of SR/MAs. Strict guidance is provided on how to summarize the critical domains and non-critical domains to determine the overall study quality.

A standardization activity was performed by three members of the research team. Six articles (10% of the included studies) were evaluated independently by each reviewer using the three checklists. The three reviewers then met to review and discuss their scores of 1–2 articles at a time before assessing the remaining included articles.

The remaining articles were assigned two reviewers (DR and ZA or NA) for independent review and data extraction according to the PRISMA-P, PRISMA, and AMSTAR-2 checklists. In the case of discrepancies between assessors, all conflicts were managed by a discussion with a third reviewer to reach consensus.

Each checklist item was given 1 mark if it was completed or 0 marks if the item was not completed. The AMSTAR-2 tool allows for "partial yes" as an assessment for some items; for analysis purposes we counted these as a full yes if there was consensus that it was not a major limitation or a full no if there were major concerns of how this could impact the interpretation of the study. The reason for this is that the guidance document does not consider partial yes assessments in the final judgement of the article (critically low, low, medium or high confidence).

For the purpose of generalizability, items which were not relevant to the study being assessed were given a full mark (for example in the PRISMA checklist: if the article was a systematic review; item 21 "present results of each meta-analysis done; including confidence intervals and measures of consistency" would be deemed not a relevant criterion, therefore, it would be given a full mark).

## Outcomes

The primary outcomes were to 1) assess the completeness of reporting of available protocols according to the PRISMA-P checklist, 2) assess the completeness of reporting of SR/MAs according to the PRISMA checklist, and 3) assess the quality of the SR/MAs according to the AMSTAR-2 tool. The secondary outcomes were 1) to determine the correlation between completeness of reporting and quality of the protocol (PRISMA-P), study (PRISMA), and the overall SR/MA quality (defined by AMSTAR-2 as critically low, low, moderate or high) and 2) describe the frequency of discrepancies in reporting between PRISMA-P and PRISMA.

A post-hoc analysis was completed after it was apparent that the rates of overall quality assessments were considered "low" or "critically low" for most articles and the reporting of protocols was poor. The AMSTAR-2 tool item 2 assesses the reporting and content of protocols and is defined as a critical item for high quality SR/MAs. We hypothesized that regardless of the presence or absence of a protocol, the results and interpretation of SR/MAs could be deemed appropriate if the other critical criteria of the AMSTAR 2 tool were met. This is to say that despite what the authors had planned, what was performed could be reasonable and adequate. We performed a sensitivity analysis assuming item 2 was given a 'yes' to determine the change in the summary assessment.

### Data synthesis

Descriptive statistics were used to summarize the results of each of the 3 checklists. The test for normality was completed using the Kolmogrov-Smirnov statistical test. The association between PRISMA-P and AMSTAR-2 and between PRISMA and AMSTAR-2, were assessed using linear regression. Multiple linear regression of all 3 checklists was completed using PRISMA and PRISMA-P scores as independent variables. Interrater reliability using Cohen's kappa coefficient statistical test and the percent agreement between raters were used to describe agreement between assessors. Statistical significance was considered with a two tailed p-value of less than 0.05. All data were analyzed using the statistical package SPSS software version 25.0 and tables and figures were configured using Microsoft Excel 2016.

## Results

### Search results

A total of 1,023 articles were identified from the initial search of the 2 databases. Upon screening 357 unique articles, 51 (14.3%) met the inclusion criteria and were included in the analysis. The most common reason for exclusion was that protocols were not available (n = 210, 68.6% of excluded articles). Fig 1 illustrates the process.

It was observed that the availability of protocols has increased steadily since 2015 (Table 1). Protocols were overwhelmingly located through registration in the PROSPERO database. PROSPERO citations were often clearly stated in the manuscript. Less common locations for protocols included the supplementary material or journal website and the bibliography as a reference to a previously published protocol. Seven articles stated that they worked from a pre-specified protocol that was only available through contacting the authors. All excluded articles, with reasons, are referenced in S2 File.

### Reporting and quality results

The average PRISMA-P score for the completeness of the included protocols was 15.7 +/- 3.3 (65.6%) with a minimum of 7 and a maximum of 23 out of a possible 24 points. Items which were not reported in more than 50% of protocols were 3b, 10, 11b, 15a, 15d, and 17. Details for each individual item of the PRISMA-P results are found in Fig 2. The summary results from each included SR/MA are described in Table 2.

Regarding the published manuscripts, adherence to the PRISMA reporting guidelines was high with an average of 25.1 ± 1.1 out of a possible 27 points (93.2%; minimum 23, maximum 27). All items were reported in more than 50% of the published manuscripts. Further details of each individual item can be found in Fig 3.

The overall quality of published manuscripts evaluated using the AMSTAR-2 tool was poor. Using the AMSTAR-2 guidance to assess the quality of the manuscripts, including

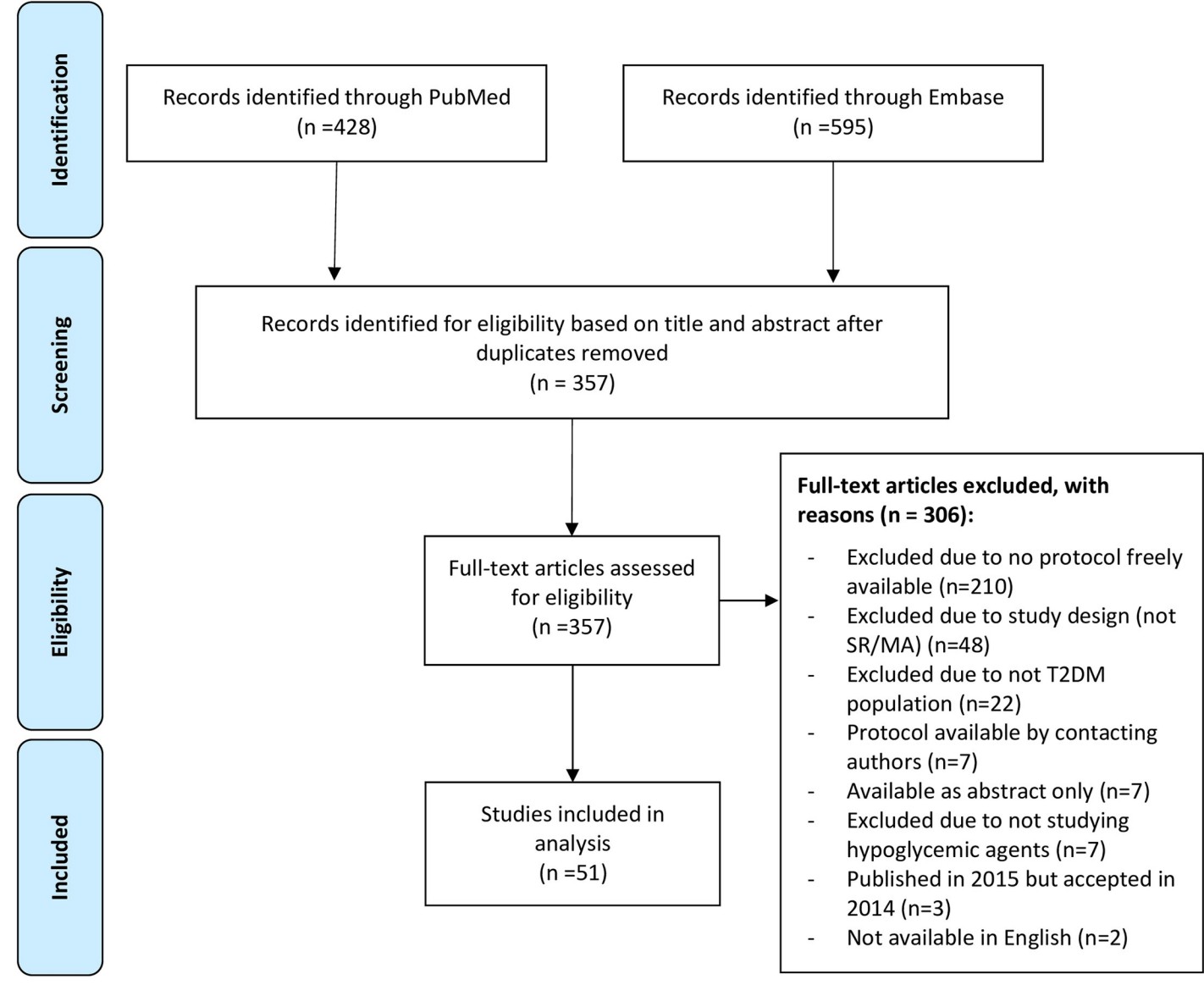

**Fig 1. Flow diagram of included studies.**

**Table 1. Characteristics of included studies (n = 51).**

| Year Published | 2015 | 2016 | 2017 | 2018 | 2019 |
|---|---|---|---|---|---|
| Protocol Available (n) | 3 | 10 | 17 | 17 | 4 |
| Met Inclusion Criteria and Protocol Not Available (n, %) | 40 (8%) | 63 (16%) | 54 (33%) | 46 (37%) | 7 (57%) |
| Protocol Access* | | | | | |
| Journal Website / Supplementary Material | 1 | 2 | 4 | 4 | 0 |
| Bibliography | 0 | 0 | 0 | 1 | 0 |
| PROSPERO | 3 | 8 | 16 | 15 | 4 |

*Some manuscripts had protocols available from multiple sources.

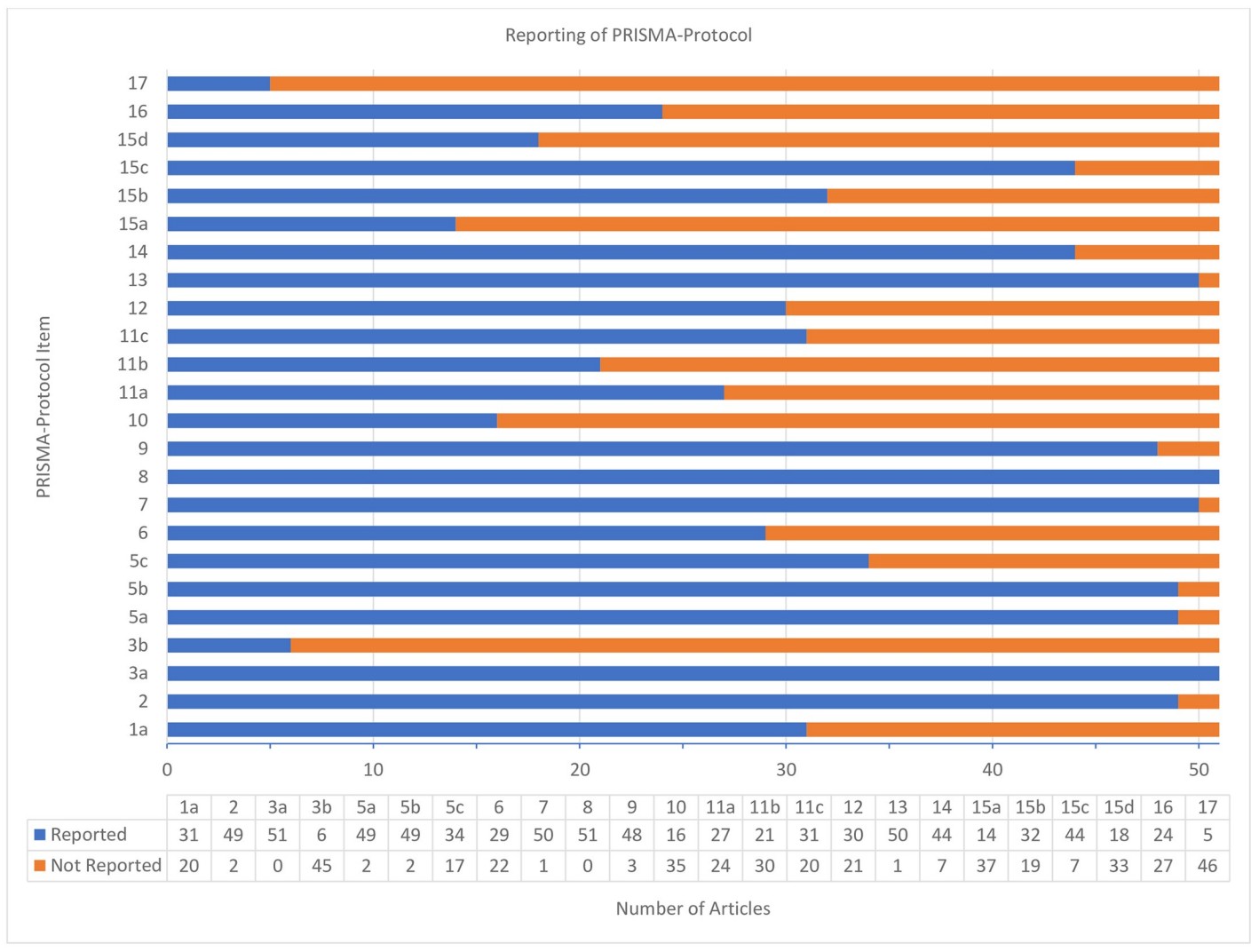

**Fig 2. Completeness of reporting of protocols as assessed by the PRISMA-P checklist.**

supplementary material, 63% were classified as critically low (n = 32), 18% as low (n = 9), 8% as moderate (n = 4), and 12% as high quality (n = 6). Further details of the critical appraisal can be found in Figs 4 and 5.

We performed a post-hoc sensitivity analysis where item number 2 (authors provided a detailed protocol) on the AMSTAR-2 tool was given a yes. The manuscripts rated as critically low decreased to 17 (33%), 20 (39%) were rated as low, 7 (14%) as moderate, and 7 (14%) as high.

## Correlations

There was no statistically significant correlation between the PRISMA-P and PRISMA (r = 0.264; p = 0.06). The strength of association was statistically significant between PRISMA-P and AMSTAR-2 (r = 0.333; $r^2$ = 0.11; p = 0.02) in addition to PRISMA and AMSTAR-2 (r = 0.555; $r^2$ = 0.31; p<0.01). While the bivariate models of PRISMA-P or PRISMA scores with AMSTAR-2 quality were statistically significant, the correlation was weak. This highlights

**Table 2. PRISMA-P, PRISMA and AMSTAR-2 quality for each included article, organized alphabetically.**

| Article | PRISMA-P Checklist Score (max = 24) | PRISMA Checklist Score (max = 27) | AMSTAR-2 Number of Critical Weaknesses† (max = 9) | AMSTAR-2 Number of Minor Weaknesses‡ (max = 7) | AMSTAR-2 Quality |
|---|---|---|---|---|---|
| Adil M Clinical Epidemiology and Global Health 2018 [15] | 17 | 25 | 2 | 3 | CL |
| Andreadis P Diabetes Obes Metab 2018 [16] | 16 | 25 | 1 | 2 | L |
| Anyanwagu U Diabetes Res Clin Pract 2016 [17] | 14 | 26 | 2 | 0 | M |
| Black CD Diabetes Therapy 2017 [18] | 23 | 27 | 0 | 0 | H |
| Cai X Diabetes Investig 2017 [19] | 18 | 24 | 3 | 2 | CL |
| Cai X Diabetes Technol Ther 2016 [20] | 18 | 23 | 2 | 6 | CL |
| Cai X Expert Opin Pharmacother 2016 [21] | 18 | 24 | 2 | 3 | CL |
| Cai X Expert Opin Pharmacother 2017 [22] | 19 | 24 | 4 | 1 | CL |
| Cai X J Diabetes Investig 2018 [23] | 18 | 25 | 1 | 1 | L |
| Cai X Obesity 2018 [24] | 17 | 23 | 2 | 1 | CL |
| Cai X PLoS ONE 2016 [25] | 17 | 25 | 3 | 2 | CL |
| Campbell JM Ageing Res Rev 2017 [26] | 18 | 27 | 0 | 4 | M |
| Castellana M Diabetes/Metabolism Research and Reviews 2019 [27] | 20 | 25 | 2 | 1 | CL |
| Crowley MJ Ann Intern Med 2017 [28] | 21 | 27 | 0 | 0 | H |
| de Wit HM Br J Clin Pharmacol 2016 [29] | 20 | 26 | 2 | 2 | CL |
| Dicembrini I Acta Diabetol 2017 [30] | 14 | 26 | 2 | 3 | CL |
| Elgebaly A Experimental and Clinical Endocrinol 2018 [31] | 16 | 26 | 1 | 1 | L |
| Elgendy IY Am J Cardiovasc Drugs 2017 [32] | 12 | 24 | 2 | 4 | CL |
| Farah D Diabetes Research and Clin Pract 2019 [33] | 14 | 22 | 4 | 4 | CL |
| Giugliano D Endocrine 2016 [34] | 10 | 24 | 3 | 3 | CL |
| Glechner A Diabetologia 2015 [35] | 14 | 26 | 2 | 2 | CL |
| Gray LJ Diabetes Obes Metab 2015 [36] | 13 | 25 | 4 | 3 | CL |
| Hansen M Diabetes Complications 2017 [37] | 18 | 26 | 0 | 2 | H |
| Khunti K Diabetes Care 2017 [38] | 11 | 26 | 2 | 4 | CL |
| Khunti K Diabetes Obes Metab 2018 [39] | 17 | 23 | 4 | 6 | CL |
| Li X Frontiers in Pharmacology 2018 [40] | 15 | 23 | 3 | 1 | CL |
| Li X Endocrine 2018 [41] | 14 | 24 | 3 | 4 | CL |
| Liao HW Endocrionology Diabetes and Metabolism 2018 [42] | 13 | 26 | 2 | 2 | CL |
| Liu X Lipids Health Dis 2016 [43] | 12 | 26 | 2 | 2 | CL |
| Maiorino MI Diabetes Care 2017 [44] | 8 | 26 | 2 | 2 | CL |

*(Continued)*

**Table 2.** (Continued)

| Article | PRISMA-P Checklist Score (max = 24) | PRISMA Checklist Score (max = 27) | AMSTAR-2 Number of Critical Weaknesses† (max = 9) | AMSTAR-2 Number of Minor Weaknesses‡ (max = 7) | AMSTAR-2 Quality |
|---|---|---|---|---|---|
| Maiorino MI Diabetes Obes Metab 2018 [45] | 7 | 22 | 2 | 4 | CL |
| Mazidi M J Am Heart Assoc 2017 [46] | 15 | 26 | 1 | 3 | L |
| Mazidi M J Diabetes Complications 2017 [47] | 13 | 25 | 1 | 2 | L |
| McGovern A Diabetes Obes Metab 2018 [48] | 23 | 26 | 2 | 2 | CL |
| Meng Q J Diabetes Investig 2016 [49] | 14 | 25 | 1 | 1 | L |
| Min SH J Diabetes Investig 2018 [50] | 15 | 24 | 2 | 3 | CL |
| Mishriky BM Diabetes and Metabolism 2018 [51] | 16 | 24 | 3 | 3 | CL |
| Monami M Acta Diabetol 2017 [52] | 14 | 25 | 2 | 2 | CL |
| Monami M Diabetes Obes Metab 2018 [53] | 15 | 25 | 2 | 3 | CL |
| Monami M Diabetes Res Clin Pract 2017 [54] | 15 | 25 | 3 | 3 | CL |
| Ostawal A Diabetes Therapy 2016 [55] | 15 | 27 | 1 | 3 | L |
| Pang B Diabetes Ther 2017 [55] | 18 | 26 | 3 | 4 | CL |
| Peter EL Ethnopharmacol 2019 [56] | 16 | 25 | 0 | 2 | M |
| Price HI BMJ Open 2015 [57] | 16 | 27 | 0 | 1 | H |
| Saad M Int J Cardiol 2017 [58] | 14 | 24 | 0 | 2 | M |
| Sharma M BMJ Open 2017 [59] | 17 | 24 | 2 | 1 | CL |
| Shi F Frontiers in Pharmacology 2018 [60] | 15 | 27 | 0 | 0 | H |
| Storgaard H PLoS One 2016 [61] | 21 | 27 | 0 | 1 | H |
| Tang GH Cancer Epidemiology Biomarkers and Prevention 2018 [62] | 18 | 25 | 1 | 1 | L |
| Wang C Diabet Obes Metab 2018 [63] | 17 | 27 | 1 | 1 | L |
| Wang X Medicine (Baltimore) 2018 [64] | 14 | 26 | 2 | 2 | CL |

†AMSTAR-2 items 1, 3, 5, 6, 8, 10, 12, 14, 16;

‡AMSTAR-2 items 2, 4, 7, 9, 11, 13, 15;

CL = critically low, L = low, M = moderate, H = high quality.

that factors outside of the completeness of reporting, and not measured in this study, contribute to these observations. Visualization of the scatterplot diagrams imply a linear relationship and can be viewed in S1 Diagrams. However, provided the smaller sample size, the clustering of PRISMA scores at the higher end with the AMSTAR-2 quality assessment being overwhelmingly critically low may threaten the linearity assumptions. When combined in multiple linear regression, there was a lack of collinearity between AMSTAR-2 quality category and the independent variables PRISMA ($p < 0.01$) and PRISMA-P ($p = 0.10$). A review of the scatterplot and the histogram of residuals (S1 Diagrams) suggest the linear regression model is reasonable.

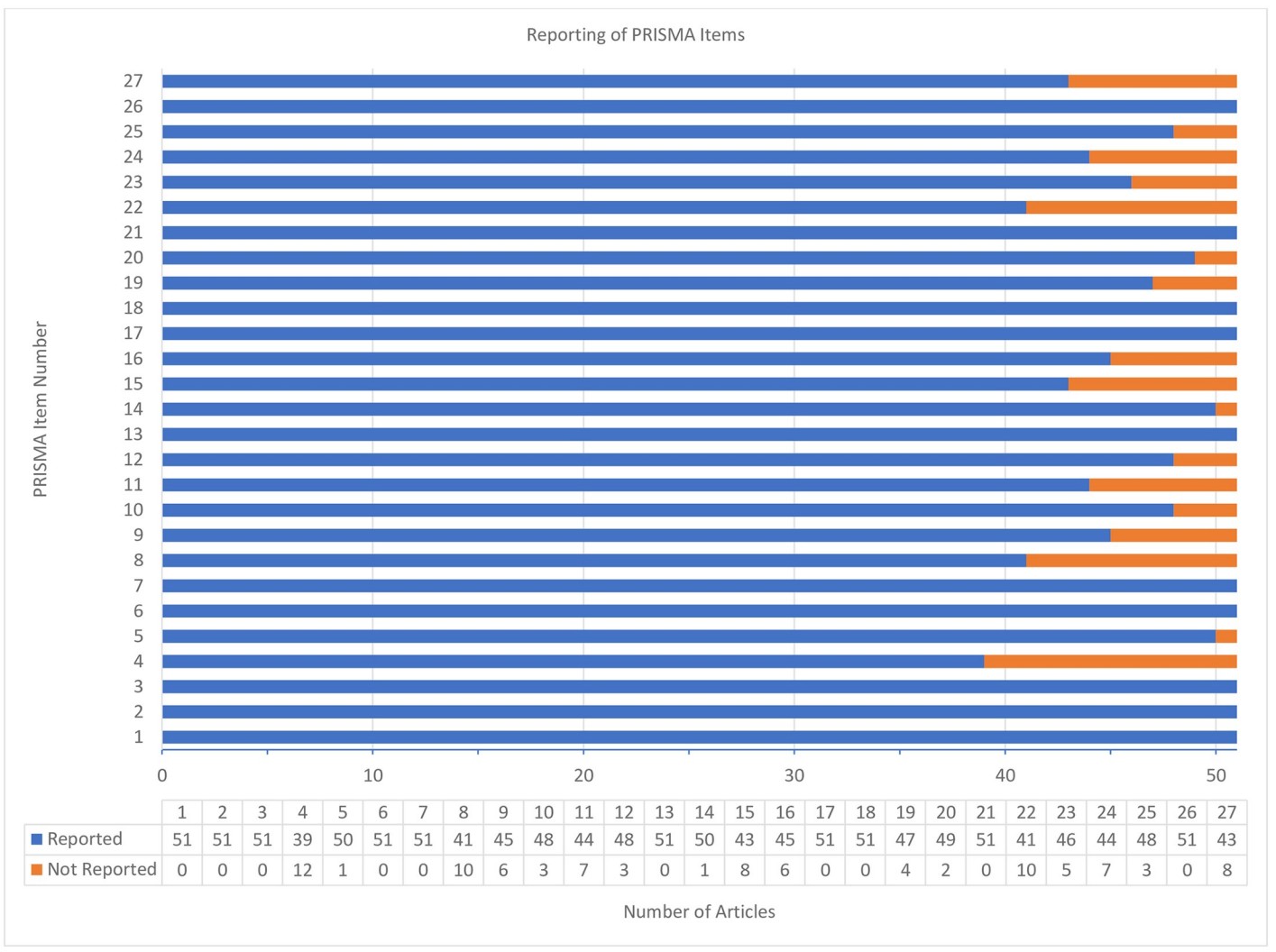

**Fig 3. Completeness of reporting of protocols as assessed by the PRISMA checklist.**

### Agreement

Interrater agreement for individual items was significant for PRISMA-P ($\kappa = 0.823$, p<0.001; percent agreement 91.9%), PRISMA ($\kappa = 0.427$, p<0.001; percent agreement 91.2%) and AMSTAR-2 ($\kappa = 0.623$, p<0.001; percent agreement 80.3%).

### Discrepancies in PRISMA and PRISMA-P reporting

The results, shown in Table 3, indicates that items reported in the manuscript according to the PRISMA checklist were not reported in the protocol according to the PRISMA-P checklist, or vice-versa, thus indicating discrepancies in reporting.

### Discussion

The results of our study show that while there is good adherence in the reporting of SR/MA manuscripts, the adherence to reporting of protocols is poor. The statistically insignificant

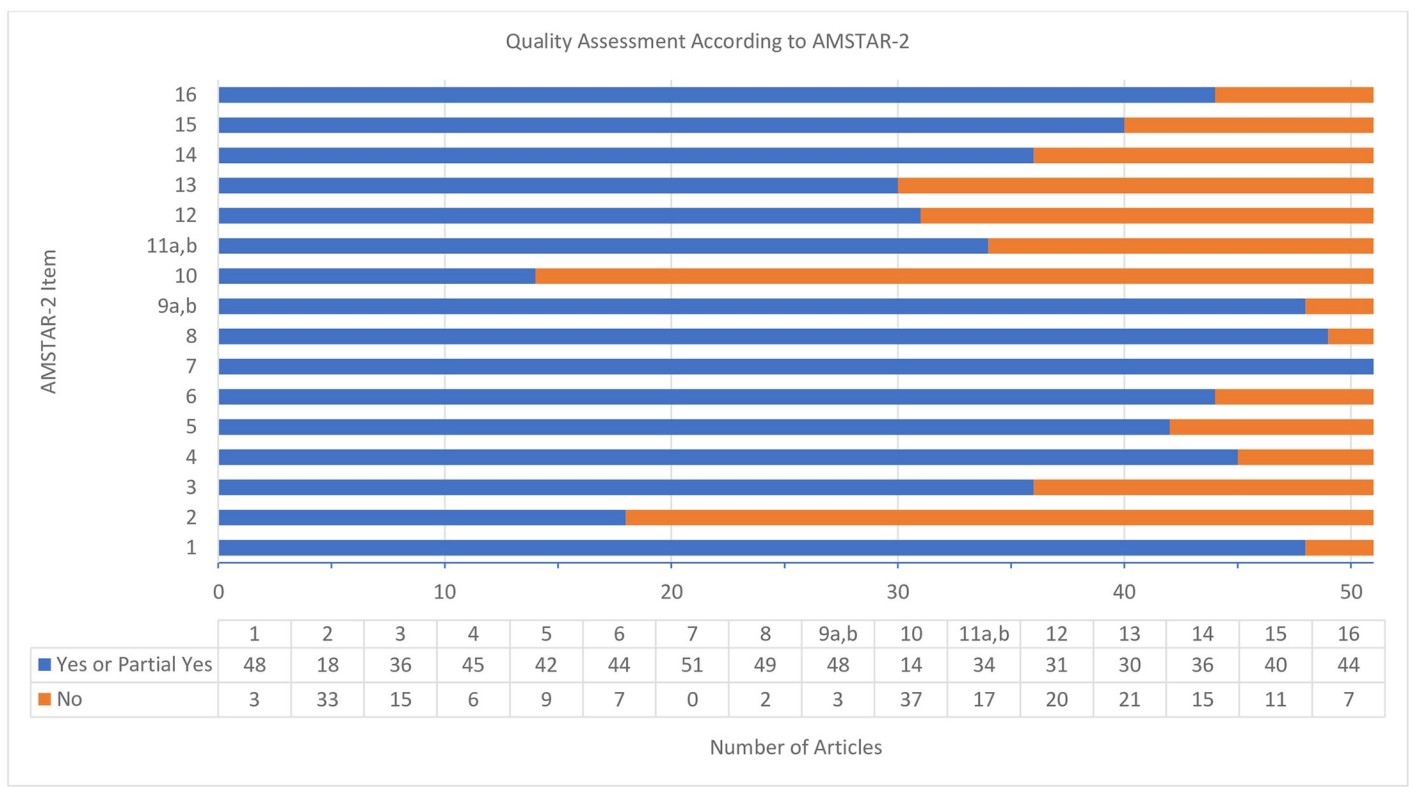

**Fig 4. Quality assessment according to AMSTAR-2 tool.**

correlation between PRISMA-P and PRISMA would suggest that if a protocol is reported well, it does not mean that the manuscript will also be reported well. Our results highlight the overall low quality of the included studies, as assessed by the AMSTAR-2 tool, and emphasizes that we must be selective when using this literature as part of evidence-based practice. Individually, higher scores of the PRISMA-P and PRISMA checklists results in an increased quality categorization, however, this correlation is weak. The existence of this correlation is intuitive as authors must provide adequate details in order to accurately assess the quality of the SR/MA. However, when adjusted, only the PRISMA score was associated with higher quality SR/MAs. This lack of collinearity identifies that other factors are responsible for increasing the quality of the evidence and is separate from reporting. These results implore users of the SR/MA literature to be thorough in their critical appraisal as the completeness of reporting of a protocol or the manuscript is an insufficient surrogate marker of quality.

Our results indicate that manuscripts were generally reported well according to the PRISMA scores. Many journals require authors to submit the PRISMA checklist along with the manuscript to demonstrate the completeness of reporting. However, some discrepancies were present when reviewing the provided checklists and performing our own assessment. Insight into the lack of correlation between the PRISMA and PRISMA-P scores in our sample of studies may be due to two possibilities: the first is due to how new the PRISMA-P statement is, published in 2015, relative to the PRISMA statement, published in 2009 (and updated from QUOROM originally published in 1999), and the unfamiliarity of what constitutes a high quality protocol. We observed that protocols are being more frequently provided and freely accessible over time exemplifying the importance of transparent research. These results are

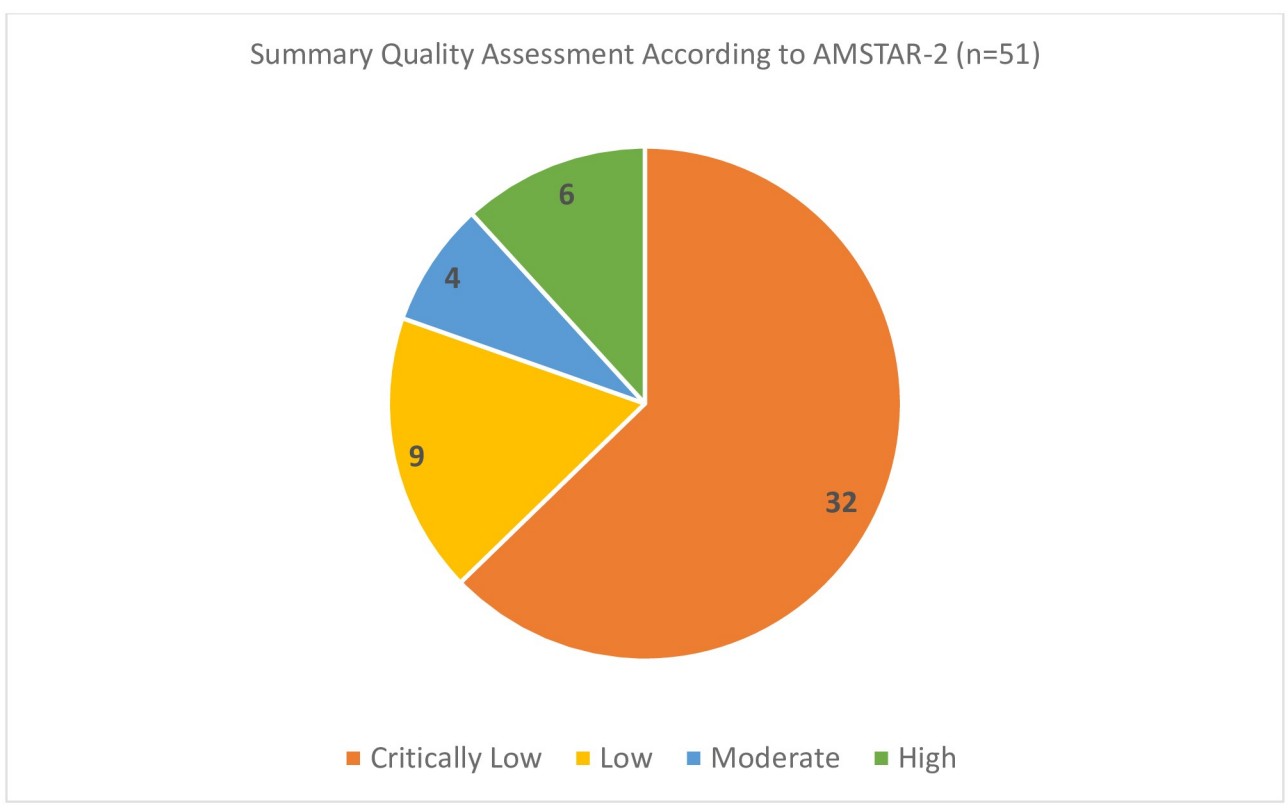

**Fig 5. Summary quality assessment according to AMSTAR-2 tool.**

consistent with the citation rates for each of these tools [65]. Given that there are several similar items between the PRISMA-P and PRISMA, the second possibility may that authors are not adequately detailing their plans prior to embarking on their data collection and analyses. This would result in either deliberate or undeliberate selective reporting bias.

**Table 3. Comparison of similar items reported in manuscripts and protocols according to PRISMA and PRISMA-P checklists.**

|  | Described in Only One Checklist (PRISMA or PRISMA-P) | Described in Both Checklists |
|---|---|---|
| **Rationale** | 22 (43%) | 29 (57%) |
| **Objectives** | 13 (25%) | 38 (75%) |
| **Eligibility** | 0 | 51 (100%) |
| **Information sources** | 3 (6%) | 48 (94%) |
| **Search strategy** | 25 (49%) | 26 (51%) |
| **Study selection** | 26 (51%) | 25 (49%) |
| **Data collection process** | 19 (37%) | 32 (63%) |
| **Data items** | 26 (51%) | 25 (49%) |
| **Risk of bias of included studies** | 10 (20%) | 41 (80%) |
| **Meta analysis methods** | 20 (39%) | 31 (61%) |
| **Additional analyses** | 31 (61%) | 20 (39%) |
| **Risk of bias across studies** | 25 (49%) | 26 (51%) |

Our results are similar to those found by Tunis and coworkers who performed a review of 130 articles from 11 of the top radiology journals and found high adherence rates to the PRISMA checklist and a strong association (r = 0.86) between PRISMA and quality assessment according to the AMSTAR tool [66]. Work by Zhang and colleagues also found a strong association between PRISMA and AMSTAR rating ($r^2$ = 0.793) in 197 surgical SR/MAs [67]. Previously, the AMSTAR tool had not prescribed a standard methodology to assess the overall quality of a SR/MA, thus common practice was to gauge the quality of a SR/MA using a total score (out of 11). The AMSTAR-2 tool highlights that not all items should be equally weighted. Each item is defined as a major or a minor criterion and uses this to provide the user clear guidance on how to assign papers to one of the four categories [3, 5]. This fundamentally different approach to defining the quality of studies could be a reason why our results found a weak association between PRISMA score and AMSTAR-2 assessed study quality, but previous work demonstrated a strong association between PRISMA score and AMSTAR score.

The methodology section of protocols contained the most common deficiencies. More than 50% of protocols did not report adequate details of the search strategy, selection process, how and if quantitative synthesis is appropriate, meta biases such as publication bias or selective reporting, and an overall summary of the evidence (e.g. using the GRADE criteria). Other deficiencies were present but were less common.

Contrary to the reporting of protocols, authors closely followed the PRISMA reporting criteria with all items being reported by at least 50% of the manuscripts. The most common missing information was an explicit PICOS statement (population, intervention, comparison, outcomes, and study design), a detailed search strategy used for at least 1 database, and the results of the risk of bias analysis across studies (e.g. publication bias or selective reporting bias).

There are several similarities in reporting items between the PRISMA and PRISMA-P checklists, although each item is found in their corresponding documents. We compared similar items to determine if the reporting was present in each document. We did not go into the details of what was provided but simply if it was present or not. For example, if the authors reported their search strategy in both the protocol and manuscript, we did not check to determine if there were discrepancies between these two information sources. We found that the most common areas for discrepancies between similar items were in the methodology of the search strategy, study selection, data items, additional analyses, and meta-biases across studies. These items are considered critical items in the AMSTAR-2 tool and directly impact the trustworthiness of the results.

The AMSTAR-2 tool identified that more than 50% of studies did not fulfill major criteria 2 (adequately described protocol) and minor criteria 10 (sources of funding). Our AMSTAR-2 results are consistent with several other published studies which indicate overall critically low or low quality SR/MAs. A review of 64 SR/MAs for pharmacological and non-pharmacological interventions in insomnia found that 40 were rated as low or critically low [68]. A review of 5 SR/MAs for acupuncture in primary dysmenorrhea found all 5 were rated as critically low [69]. As of October 2019, the PROSPERO database no longer accepts protocols in which data extraction has already started. Users of the literature should expect to see new SR/MA protocols published as supplementary material. This could be a possible methodological concern as authors could change the protocol before publishing without a clear audit trail unless protocols are meticulously documented. Our study did not differentiate between those registered or updated in the PROSPERO database, those available in the supplementary material, or those which were previously published.

Our study is not without its limitations. First, our search strategy used type 2 diabetes as a MeSH term while using it as a keyword may have produced more results to evaluate, thus potentially limiting the generalizability of the study. We also limited our results to English

which may have decreased our available sample. Second, we did not contact study authors for their protocol which may have increased the sample size. However, our approach follows a practical approach that most users of the literature would follow. Third, our assessment of overall SR/MA study quality may be an overestimate as all items counted as "partial yes" were considered a full yes. We acknowledge our only deviation from the AMSTAR-2 tool which was on item 7 where if a clear, complete PRISMA flow diagram was included, we gave the authors a partial yes. Only 3 articles would have met the criteria for full yes. Finally, despite high interrater agreement ranging from 80% to over 90%, lower Kappa values were found. Lower Kappa values are common for ratings which are binary and have an uneven distribution (e.g. 95% yes, 5% no). The risk of agreement due to chance alone is not accurate in this scenario and should not be interpreted in isolation.

## Conclusion

Since the inception of the PRISMA-Protocol in 2015, there has been a steady increase in the number of available SR/MA protocols over the years. However, still less than 1 in 3 SR/MAs have available protocols. Even if protocols are available, the average rate of completion is 65.6% according to reporting guidelines. The quality of recently published SR/MAs are surprisingly poor, even when disregarding the quality of protocols. Journals should encourage authors to follow PRISMA-P guidance as closely as authors currently follow PRISMA reporting guidelines. The most critical areas for improvement are in the details of the provided protocol, investigating the causes of heterogeneity, and the impact of risk of bias on the evidence synthesis.

This study should serve as an alert to both authors and users of the medical literature. The risk of using low or critically low SR/MAs in the decision-making process for patients with T2DM is surprisingly high. Users should take care to critically appraise articles to assess the reliability and accuracy of SR/MAs before applying those results to their clinical practice. Authors should be aware of how their research will be assessed and prepare their research appropriately by incorporating clear and complete protocols, along with their typically well reported manuscripts. It is clear that there are still many areas for improvements in the literature that is currently available. Authors should start with the development of a protocol prior to embarking on a systematic review.

## Supporting information

**S1 File.**
(PDF)

**S2 File.**
(DOCX)

**S1 Diagrams.**
(DOCX)

**S1 Checklist. PRISMA 2009 checklist.**
(PDF)

## Author Contributions

**Conceptualization:** Daniel Christopher Rainkie, Zeinab Salman Abedini, Nada Nabil Abdelkader.

**Data curation:** Daniel Christopher Rainkie, Zeinab Salman Abedini, Nada Nabil Abdelkader.

**Formal analysis:** Daniel Christopher Rainkie, Zeinab Salman Abedini, Nada Nabil Abdelkader.

**Funding acquisition:** Daniel Christopher Rainkie.

**Investigation:** Daniel Christopher Rainkie, Zeinab Salman Abedini, Nada Nabil Abdelkader.

**Methodology:** Daniel Christopher Rainkie, Zeinab Salman Abedini, Nada Nabil Abdelkader.

**Project administration:** Daniel Christopher Rainkie, Zeinab Salman Abedini, Nada Nabil Abdelkader.

**Resources:** Daniel Christopher Rainkie, Zeinab Salman Abedini, Nada Nabil Abdelkader.

**Software:** Daniel Christopher Rainkie, Zeinab Salman Abedini, Nada Nabil Abdelkader.

**Supervision:** Daniel Christopher Rainkie.

**Validation:** Daniel Christopher Rainkie, Zeinab Salman Abedini, Nada Nabil Abdelkader.

**Visualization:** Daniel Christopher Rainkie, Zeinab Salman Abedini, Nada Nabil Abdelkader.

**Writing – original draft:** Daniel Christopher Rainkie, Zeinab Salman Abedini, Nada Nabil Abdelkader.

**Writing – review & editing:** Daniel Christopher Rainkie.

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
