## [Decision Letter · Decision Letter 0]

10 Aug 2020

PONE-D-20-16992

Reporting and Methodological Quality of Systematic Reviews and Meta-Analysis with Protocols in Diabetes Mellitus Type II: A Systematic Review

PLOS ONE

Dear Dr. Rainkie,

Thank you for submitting your manuscript to PLOS ONE. After careful consideration, we feel that it has merit but does not fully meet PLOS ONE’s publication criteria as it currently stands. Therefore, we invite you to submit a revised version of the manuscript that addresses the points raised during the review process.

We look forward to receiving your revised manuscript.

Kind regards,

Ahmed Negida, MD

Academic Editor

PLOS ONE

Journal Requirements:

Reviewers' comments:

Reviewer's Responses to Questions

**Comments to the Author**

1. Is the manuscript technically sound, and do the data support the conclusions?

Reviewer #1: Yes

Reviewer #2: Yes

Reviewer #3: Yes

2. Has the statistical analysis been performed appropriately and rigorously? 

Reviewer #1: Yes

Reviewer #2: N/A

Reviewer #3: Yes

3. Have the authors made all data underlying the findings in their manuscript fully available?

Reviewer #1: Yes

Reviewer #2: Yes

Reviewer #3: Yes

4. Is the manuscript presented in an intelligible fashion and written in standard English?

Reviewer #1: No

Reviewer #2: Yes

Reviewer #3: No

5. Review Comments to the Author

Reviewer #1: In this review, the authors assessed the quality of published SRs in the domain of type 2 diabetes treatments. They did this by assessing fulfillment of the PRISMA, PRISMA-P and AMSTAR2 checklists. Given the importance and prevalence of diabetes as well as the emerging treatment options for this condition, and the fact that SRs are generally placed at the top of the evidence hierarchy, an evaluation of the quality of published reviews is an excellent addition to the existing body of literature, especially given the relative recency of the AMSTAR2 checklist. Overall, the manuscript is quite well-written and the authors seem to have done a thorough job with the review; nevertheless, there are a few recommendations I’ve attached below that I believe would further improve the quality of what is an already impressive manuscript. All comments were attached in the Word file.

Reviewer #2: This is a systematic review in which the authors investiagte the reporting and methodological quality of the different systematic reviews, meta-analyses, and protocols related to Type 2 DM. Although the study is well-written, well-structured, and scientifically-sound, many other literatures investigated the same topic with similar results. Despite the good quality of the manuscript, it lacks the novelity to be published in PLOS ONE.

Reviewer #3: The authors performed this study to evaluate (1) the completeness of reporting of available protocols according to the PRISMA-P checklist, (2) the completeness of reporting of SR/MAs according to the PRISMA checklist, and (3) the quality of the SR/AMs according to the AMSTAR-2 tool.

Although the overall approach to the review is proper, I have a few comments:

1. General

- There is an improper use of abbreviations in both Abstract and Manuscript file. Abbreviations should be defined at first mention and used consistently thereafter.

2. Abstract

- Objectives are not clearly outlined, please modify.

3. Methods

- Search Strategy: using PubMed filters, such as Human, is not preferred, because some relevant articles may have been missed.

4. Results

- Search Results are inconsistent with the numbers in Figure 1. For example, Line 178-180: 52 studies met the inclusion criteria; however, it is "51" in Figure 1. Also, protocols were not available in 209 studies; however, it is 203 in Figure 1. Please recheck!!

5. Language: The entire manuscript needs extensive professional revision for grammatical errors and stylistic editing to improve the quality of English. For example,

- Line 20-22: Consider using a comma instead of a semicolon.

- Line 28, 33, etc.: When the sentence contains a series of three or more words, phrases, or clauses. Consider inserting a comma to separate the elements.

- Line 55: replace “to produce” with “in producing”

- Line 57: replace “to evaluating” with “to evaluate”

- Line 67-68: “for completing a SR/MA for transparent science and to minimize”, Faulty parallelism.

6. PLOS authors have the option to publish the peer review history of their article (what does this mean?). If published, this will include your full peer review and any attached files.

Reviewer #1: No

Reviewer #2: No

Reviewer #3: No

---

## [Author Response · Author response to Decision Letter 0]

1 Sep 2020

Dear Dr Negida, Peer Reviewers, and the editorial team,

On behalf of my research team, we would like to thank you all for the time and effort you have made to review our manuscript titled “Reporting and Methodological Quality of Systematic Reviews and Meta-Analysis with Protocols in Diabetes Mellitus Type II: A Systematic Review”. We would like to offer a special thanks to those who provided detailed feedback for how to further improve our manuscript. 

Below you will find our responses to each comment and their associated actions written in blue text. We hope that this will meet your expectations.

Review Comments to the Author

Reviewer 1

We thank the reviewer for their attention to detail and feedback provided to us. We appreciate the time that was spent on their detailed report. This provides a great model feedback to those students involved in this project. We thank you for your kind words regarding the quality of this study. We spent many hours reviewing and revising this manuscript prior to the submission.

• General Advice:

o Periods should be placed after -not before- citations.

Thank you for highlighting this. We have revised the entire manuscript and double checked that all citations are provided before the end of each sentence. These updates have not been highlighted in the track changes of the manuscript.

o Although generally well-written, the manuscript would probably benefit from a revision of the grammar throughout. I have pointed out some examples of linguistic errors below, but there are a few others I haven’t included below.

We thank the review for their attention to detail regarding this matter. In addition to the comments provided below, we have also applied the Microsoft Editor, Grammarly, and an independent third party to review the grammar of the manuscript.

Major Revisions:

• Intro

o Page 5, line 77: Linguistic error “The secondary outcomes for this study was to”

Was updated to were

• Methods:

o Page 6, line 92 and 95: PubMed is not a database, it is a search engine designed to browse MEDLINE, which itself is the database.

Thank you for this correct. We have updated the sentence to read the following: “A systematic search was performed across MEDLINE (using PubMed) and Embase databases…”

o Page 6, line 96 to 100: The search strategy does not seem very comprehensive (only two terms for diabetes were used), which may be a concerning limitation as it may limit the representativeness of the retrieved SRs. This is addressed in the limitations section of the discussion, and I commend the authors for acknowledging this point.

We thank you for this comment and agree that this is indeed a limitation and is important to note. In the development of our protocol we debated the use of using keywords or the MeSH term for type 2 diabetes. We finalized our choice on the MeSH term to make the search result more specific. On review, using keywords for type 2 diabetes instead of the MeSH term we found that the number of results does not increase substantially. Repeating the search on Aug 24, 2020, we found that the search results increase from 428 to 469. If we extrapolate our inclusion/exclusion criteria results to these 41 articles, we may have included an additional 5-6 articles. 

o Eligibility Criteria:

Page 7, line 113-114: “meta-analysis” should be changed to “analyses” (Plural)

• Thank you, this has been corrected. The sentence now reads, “Other types of excluded studies included guidelines, select article meta-analyses (i.e. no systematic review), network meta-analyses, and individual patient meta-analyses because the issues specific to each of these designs are not addressed in the PRISMA-P, PRISMA or AMSTAR2 tools”.

o Study Selection:

Page 7, line 117. Linguistic error: “By independently by”

• Thank you, we have removed the first “by” to now read, “The selection of the studies was completed independently by all three reviewers NA, ZA, and DR”.

I believe clarifying two points in the screening process would benefit the manuscript. 

• First, were all records identified screened by all 3 reviewers (IE triple screening?)

o This is correct, there was triple screening. We have updated the sentence as noted above.

• Second, how were conflicts settled upon? (Discussion? Referral to an independent individual? (Obviously not a fourth author since the manuscript only has three authors)

o Thank you for identifying this. We have added 2 sentences to clarify the screening process as follows: “If two or more reviewers identified an article for inclusion on screening, the full text was retrieved. If only a single reviewer identified an article for inclusion on screening, a final decision to retrieve the full text was based on discussion between all the reviewers.”

o Data Management & Extraction:

Page 7, line 131: Linguistic error: “Checklist to the 1-2 articles at a time”

• This has been updated to read the following: “All three reviewers independently applied the checklists to 1-2 articles at a time.”

Page 8, line 145: Linguistic error: “items which were not relevant to the study being assessed were given we gave the authors a full mark”

• This has been updated to read the following: “For the purpose of generalizability, items which were not relevant to the study being assessed were given a full mark”

Page 8, line 147-148: Linguistic error: “Given full mark” to “Given a full mark”

• This has been updated in the text. 

o Outcomes:

Line 154-155: Linguistic error: “Correlation between….compared” to “Correlation between….and”

• Thank you for noting this. We have updated the sentence to read: “The secondary outcomes were to determine the correlation between completeness of reporting and quality of the protocol (PRISMA-P), study (PRISMA), and the overall SR/MA quality (defined by AMSTAR-2 as critically low, low, moderate or high) and describe the frequency of discrepancies in reporting between PRISMA-P and PRISMA.”

Line 157: Linguistic error: “After it was it was”

• We have removed the duplication

Line 159-160: Change to “Hypothesized that regardless of the absence or presence” (Rather than “Despite the absence or presence”)

• Thank you for this, we have updated the sentence to read: “On review, we hypothesized that regardless of the presence or absence of a protocol, the results and interpretation of SR/MAs could be deemed appropriate if the other critical criteria of the AMSTAR 2 tool were met.”

o Data Synthesis:

Line 170: I believe it would benefit the clarity of the manuscript if the authors were to clarify what variables were entered into the multiple linear regression model (Presumably they entered PRISMA and PRISMA-P as the independent variables, but this should be made sufficiently clear to the readers). 

• We thank you for this comment and you are correct. We have added an additional sentence to clarify this point “PRISMA and PRISMA-P scores were used as independent variables.”

o Results

Search Results

• Page 10, Line 191: Change “7” to “seven”.

o This has been updated in the text.

• The stated number of included studies is 52; however, the PRISMA diagram shows 51 included studies. It is probably a slight error, but it should be corrected both for transparency’s sake and as it affects the calculation of proportions/percentages (See later)

• The percentage of included studies given is 14.6% (which corresponds to 51 in the PRISMA diagram, but does not correspond to 52 (14.9%) as written in the Results section)

o Thank you for your detailed review and catching this typo. The results text has been updated to 51 and now matches the PRISMA flow diagram shown in Figure 1.

Reporting and Quality Results

• Page 11, line 198: “Items in which” to “Items which”

o This has been updated in the text.

• Line 214 to 217: I believe the authors may have made a slight mistake when calculating the percentages. The stated total number of included articles is 52. The number of articles of critically low quality is 32. The stated percentage is 63%, while the actual percentage (32/52) is 61.5%. I believe the same error is present in the sensitivity analysis. This may be due to the aforementioned error in stating the number of included studies.

o Thank you for your detailed review of our calculations. With the result number corrected to 51 studies, the percentages now match. 

Correlations:

• Page 17, line 230-231: I believe the phrase “Can be used to independently predict” should be removed, as it implies a stronger correlation than there really is.

o Thank you for this comment. We agree that reading this part of the sentence by itself is misleading. We have updated the sentence to read the following: “While the independent scores of PRISMA-P and PRISMA could both be used to predict AMSTAR2 quality category, the correlation was weak.”

• Line 233: I believe the manuscript would benefit if the authors clarified what variables were entered into the multiple linear regression model. Was it a bivariate model where PRISMA and PRISMA-P were the independent variables? If so, the authors should clarify that the correlation score between PRISMA and PRISMA-P suggested a lack of collinearity and therefore justified entering both variables into the same model, as most readers would (wrongly, as it turns out) assume a high degree of collinearity between these two scores.

o Thank you for this comment. We agree that this should be detailed out in the text. We have updated the sentence to read the following: “However, when combined in multiple linear regression, there was a lack of collinearity between AMSTAR2 quality category and the independent variables PRISMA (p<0.01) and PRISMA-P (p=0.10).”

• I believe it the reliability of the manuscript would be improved if the authors clarified how the linearity assumption was verified (e.g. By providing a scatter plot), as well as, in the case of the multiple linear model, the assumptions of homoscedasticity and the normal distribution of the residuals.

o We thank the reviewer for this comment. We have included the scatter plots for each of the bivariate and multivariate analyses in Supplementary material 4. We have included the following modifications to further satisfy the linearity assumption: While the bivariate models of PRISMA-P or PRISMA scores with AMSTAR2 quality were statistically significant, the correlation was weak. This highlights that other factors are responsible rather than completeness of reporting. Visualization of the scatterplot diagrams imply a possible linear relationship and can be viewed in Supplementary File 4. However, provided the smaller sample size, the clustering of PRISMA scores at the higher end with the AMSTAR2 quality assessment being overwhelmingly critically low may threaten the linearity assumptions. When combined in multiple linear regression, there was a lack of collinearity between AMSTAR2 quality category and the independent variables PRISMA (p<0.01) and PRISMA-P (p=0.10). A review of the scatterplot and the histogram of residuals (Supplementary File 4) suggest the linear regression model is reasonable.

Discrepancies in PRISMA and PRISMA-P Reporting

• Table 3: “Only 1” to “Only one”

o This has been updated in the text.

• Table 3: Numbers add up to 51 (As stated in the PRISMA diagram), not 52 (As stated in the Methods section)

o This has been addressed in our previous updates.

• Table 3: I believe the quality of the manuscript would be improved by including the %s next to the raw numbers, as that would facilitate the reading and interpretation of the table.

o We thank you for this comment. We agree that this would help readers understand this table easier to understand. All numbers in this table have been updated using the total number of included studies as 51.

o Discussion

Line 250-251: “Lack of correlation between PRISMA and PRISMA-P”

• I believe the authors should not emphasize the “lack of correlation”, as a P-value of greater than 0.05 does not definitively prove the lack of a correlation, merely the absence of sufficient evidence for said correlation. Instead, the authors should emphasize a tone of statistical insignificance rather than a definitive lack. They should also emphasize the weakness of the correlation (even if it were statistically significant) to better deliver home the point.

o Thank you for your comment. We agree with this statement and that it is important to drive home that even if this were to be a type 2 error, it is possible that the correlation will remain weak. We have updated these sentences to read the following: “The statistically insignificant correlation between PRISMA-P and PRISMA would suggest that if a protocol is reported well, it does not mean that the manuscript will also be reported well. This result, along with the weakness of the correlation, implore that users of the literature must look at these aspects independently.”

• I believe exploring potential reasons for the lack of correlation would improve the quality of the paper. For instance, the authors state that the dates of publication of the PRISMA and PRISMA-P may cause the lack of said correlation. Could other factors, such as a difference in points covered by either checklist, account for the lack of correlation? Another implication may be that authors simply do not invest equivalent amounts of time and/or effort into fulfilling both checklists.

o We thank you for this addition to the discussion. Provided that there are many similarities between the PRISMA-P and PRISMA checklists (table 3), it is our hypothesis that it is either that authors do not spend adequate time detailing their protocol, thus potentially leading to selective reporting bias, or that authors are unfamiliar with the expectations of a high quality protocol. We have updated the first and second paragraphs of our discussion to read the following: 

The results of our study show that while there is good adherence in the reporting of SR/MA manuscripts, the adherence to reporting of protocols is poor. The statistically insignificant correlation between PRISMA-P and PRISMA would suggest that if a protocol is reported well, it does not mean that the manuscript will also be reported well. This result, along with the weakness of the correlation, implore that users of the literature must look at each of these aspects independently. Our results highlight the overall low quality of the included studies, as assessed by the AMSTAR-2 tool, emphasizes that we must be selective when using this literature as part of evidence-based practice. Individually, higher scores of the PRISMA-P and PRISMA checklists results in an increased quality categorization, however, this correlation is weak. The existence of this correlation is intuitive as authors must provide adequate details in order to accurately assess the quality of the SR/MA. However, when adjusted, only the PRISMA score was associated with higher quality SR/MAs. This lack of collinearity identifies that other factors are responsible for increasing the quality of the evidence and is separate from reporting. 

o Our results indicate that manuscripts were generally reported well according to the PRISMA scores. Many journals require authors to submit the PRISMA checklist along with the manuscript to demonstrate the completeness of reporting. However, some discrepancies were present when reviewing the provided checklists and performing our own assessment. Insight into the lack of correlation between the PRISMA and PRISMA-P scores in our sample of studies may be due to two possibilities: the first is due to how new the PRISMA-P statement is, published in 2015, relative to the PRISMA statement, published in 2009 (and updated from QUOROM originally published in 1999), and the unfamiliarity of what constitutes a high quality protocol. We observed that protocols are being more frequently provided and freely accessible over time exemplifying the importance of transparent research. These results are consistent with the citation rates for each of these tools [65]. Given that there are several similar items between the PRISMA-P and PRISMA, the second possibility may that authors are not adequately detailing their plans prior to embarking on their data collection and analyses. This would result in either deliberate or undeliberate selective reporting bias

Line 258-259: Linguistic error: “The higher the score of PRISMA-P and PRISMA tools results in a higher quality categorization according to the AMSTAR-2 tool”

• Thank you. We have updated the first paragraph to hopefully be clearer.

Line 260: Should be changed to “The presence of a correlation is intuitively correct”, as it is not intuitive that the correlation would be relatively weak as found by the authors (But the existence of a statistically significant correlation would be expected).

• We thank you for this comment and certainly agree with this. We have updated the sentence to read: “The existence of this correlation is intuitive as authors must provide adequate details in order to accurately assess the quality of the SR/MA.”

Starting from line 289, the point about limiting discrepancies to the mere presence/absence of details in the protocol/manuscript rather than actually checking for differences in the details provided is a major limitation. This is because many areas, such as statistical analysis, may be significantly modified between the protocol and the original study. The mere fact that they’re present both in the manuscript and the protocol would not warrant a “No discrepancy” judgement. Nevertheless, I commend the authors for acknowledging this limitation.

• We thank the reviewer for their comment. We agree that this is important for readers to understand. While this was outside of the intended scope of this project, we are conducting additional research on these discrepancies and the risk of selective reporting bias. 

Line 320-321: Regarding the point about Kappa coefficients, it would be more accurate to state that low kappa ratings are common when responses to a binary variable fall within a single category (e.g. 95% Yes 5% No), had the binary variable answers been more evenly distributed (e.g. 50% Yes 50% No) the kappa coefficient would have been more reasonable. It is not simply that binary variables give a low kappa coefficient as implied in lines 320-321. 

• We thank the reviewer for this comment. We have clarified the last sentence of the limitations to read the following: 

“Finally, despite high interrater agreement ranging from 80% to over 90%, lower Kappa values were found. Lower Kappa values are common for ratings which are binary and have an uneven distribution (e.g. 95% yes, 5% no). The risk of agreement due to chance alone is not accurate in this scenario and should not be interpreted in isolation.’

Reviewer 2

This is a systematic review in which the authors investiagte the reporting and methodological quality of

the different systematic reviews, meta-analyses, and protocols related to Type 2 DM. Although the study is well-written, well-structured, and scientifically-sound, many other literatures investigated the same topic with similar results. Despite the good quality of the manuscript, it lacks the novelity to be published in PLOS ONE.

We would like to thank the reviewer for taking the time to review our manuscript and their positive comments. We believe that this research will provide readers a better understanding of the interplay between reporting checklists and the quality of articles. There are some key disparities between AMSTAR and AMSTAR-2 which are highlighted here. Additionally, there are few studies which advance the understanding of the PRISMA extensions. We believe that our study promotes a better understanding of how PRISMA-P is used or not used, and where this science can help improve the quality of the literature being produced. 

Reviewer 3

The authors performed this study to evaluate (1) the completeness of reporting of available protocols

according to the PRISMA-P checklist, (2) the completeness of reporting of SR/MAs according to the PRISMA checklist, and (3) the quality of the SR/AMs according to the AMSTAR-2 tool.

Although the overall approach to the review is proper, I have a few comments:

1. General

- There is an improper use of abbreviations in both Abstract and Manuscript file. Abbreviations should be defined at first mention and used consistently thereafter.

We thank the reviewer for their comments. We have reviewed the text in full and identified the first case of abbreviation that is used and used the said abbreviation in the remainder of the manuscript. Changes have been made throughout. 

2. Abstract

- Objectives are not clearly outlined, please modify.

We thank the author for their comment. Due to the word limit, we have attempted to summarize the objectives of our work in as few words as possible. We have updated the objectives to read the following: “The objectives of this study were to describe the completeness of reporting and quality of SR/SRMAs, assess the correlations between PRISMA-P, PRISMA and AMSTAR-2 and identify reporting characteristics between similar items of PRISMA-P and PRISMA.”

3. Methods

- Search Strategy: using PubMed filters, such as Human, is not preferred, because some relevant articles may have been missed.

We thank the reviewer for this comment. On review of excluding Human as a search limit, it increased the number of articles by 2. 

4. Results

- Search Results are inconsistent with the numbers in Figure 1. For example, Line 178-180: 52 studies met the inclusion criteria; however, it is "51" in Figure 1. Also, protocols were not available in 209 studies; however, it is 203 in Figure 1. Please recheck!!

We thank the reviewer for their time in taking a detailed look at our manuscript. We have fixed the typo in the number of included studies and is now correct at 51 included articles. On detailed review of Figure 1 and our database, we have also identified 3 other unintentional errors. The number of excluded articles due to no protocol being freely available = 210, the total number of full text articles missed the addition of “available as abstract” (n=7), and protocols available by contacting authors should be 7, not 8. Thus bringing the total number of excluded articles to 306. The entirety of Figure 1 has been updated. The first paragraph of the results section has been updated to reflect this change. The abstract has also been updated. We have double checked these results with our database and the supplementary material provided. We can confidently say that these numbers are now accurate.

5. Language: The entire manuscript needs extensive professional revision for grammatical errors and stylistic editing to improve the quality of English. For example,

- Line 20-22: Consider using a comma instead of a semicolon.

o Thank you for your comment, we have updated the Objectives section of the abstract to be clearer.

- Line 28, 33, etc.: When the sentence contains a series of three or more words, phrases, or clauses. Consider inserting a comma to separate the elements.

o Thank you for your comment. We have updated the grammar of the Methods section of the abstract to be clearer.

- Line 55: replace “to produce” with “in producing”

o This has been updated in the text.

- Line 57: replace “to evaluating” with “to evaluate”

o This has been updated in the text.

- Line 67-68: “for completing a SR/MA for transparent science and to minimize”, Faulty parallelism.

o Thank you for your comment. We have changed the word “minimize” to “identify” to clarify the intended purpose of this sentence. It now reads as the following: “It is recommended that a pre-specified protocol should be used as the foundation for completing a SR/MA for transparent science and identify the risk of selection and reporting bias.”

Once again we would like to take this opportunity to thank the reviewers for their detailed remarks. We hope that this manuscript is now meets the high quality of science that is expected in PLOS ONE. 

Sincerely,

Dr Daniel Rainkie, on behalf of the investigators

---

## [Decision Letter · Decision Letter 1]

30 Sep 2020

PONE-D-20-16992R1

Reporting and Methodological Quality of Systematic Reviews and Meta-Analysis with Protocols in Diabetes Mellitus Type II: A Systematic Review

PLOS ONE

Dear Dr. Rainkie,

Thank you for submitting your manuscript to PLOS ONE. After careful consideration, we feel that it has merit but does not fully meet PLOS ONE’s publication criteria as it currently stands. Therefore, we invite you to submit a revised version of the manuscript that addresses the points raised during the review process.

ACADEMIC EDITOR: Please insert comments here and delete this placeholder text when finished. Be sure to:

Indicate which changes you require for acceptance versus which changes you recommendAddress any conflicts between the reviews so that it's clear which advice the authors should followProvide specific feedback from your evaluation of the manuscript

We look forward to receiving your revised manuscript.

Kind regards,

Hazel Bautista

Academic Editor

PLOS ONE

Journal Requirements:

Additional Editor Comments (if provided):

Your submission requires substantial editing for English grammar and usage. We ask that you please have the manuscript copyedited by either a colleague or a professional copy-editing service. While you may approach any qualified individual or any professional scientific editing service of your choice, PLOS has partnered with American Journal Experts (AJE) to provide discounted services to PLOS authors. AJE has extensive experience helping authors meet PLOS guidelines and can provide language editing, translation, manuscript formatting, and figure formatting to ensure your manuscript meets our submission guidelines. If the PLOS editorial team finds any language issues in text that AJE has edited, AJE will re-edit the text for free. To take advantage of this special partnership, use the following link: https://www.aje.com/go/plos/.

Reviewers' comments:

Reviewer's Responses to Questions

**Comments to the Author**

1. If the authors have adequately addressed your comments raised in a previous round of review and you feel that this manuscript is now acceptable for publication, you may indicate that here to bypass the “Comments to the Author” section, enter your conflict of interest statement in the “Confidential to Editor” section, and submit your "Accept" recommendation.

Reviewer #1: All comments have been addressed

Reviewer #3: All comments have been addressed

2. Is the manuscript technically sound, and do the data support the conclusions?

Reviewer #1: Yes

Reviewer #3: Yes

3. Has the statistical analysis been performed appropriately and rigorously? 

Reviewer #1: Yes

Reviewer #3: Yes

4. Have the authors made all data underlying the findings in their manuscript fully available?

Reviewer #1: Yes

Reviewer #3: Yes

5. Is the manuscript presented in an intelligible fashion and written in standard English?

Reviewer #1: Yes

Reviewer #3: Yes

6. Review Comments to the Author

Reviewer #1: The authors addressed all my comments and this version of the manuscript is well-written, well-presented, and support their conclusion. I recommend the acceptance of this manuscript.

Reviewer #3: The authors have addressed all my comments/suggestions. I found their responses quite satisfactory and the revised version has been much improved.

7. PLOS authors have the option to publish the peer review history of their article (what does this mean?). If published, this will include your full peer review and any attached files.

Reviewer #1: No

Reviewer #3: No

---

## [Author Response · Author response to Decision Letter 1]

29 Oct 2020

Dear Dr Negida, Peer Reviewers, and the editorial team,

On behalf of my research team, we would like to thank you all for the time and effort you have made to once again review our manuscript titled “Reporting and Methodological Quality of Systematic Reviews and Meta-Analysis with Protocols in Diabetes Mellitus Type II: A Systematic Review”. We are very pleased that we have addressed all of your comments regarding the scientific merit of our manuscript in our first revision. We thank you for the opportunity to continue to improve our work with a thorough review of grammar and English usage. We have spent considerable time and effort revising the manuscript for grammatical errors and wording. 

In our efforts, we have had two colleagues review this manuscript. The first was Dr Zachariah Nazar, an assistant professor with a PhD in pharmacy and a native English speaker. He identified many areas for improvement regarding our English use in our manuscript. The second was Ms Kara Schultz, an English teacher with a Masters of Arts in Language Studies and a native English speaker. She further identified stylistic and grammar errors. The entire manuscript has been reviewed once again in full by the research team, one native English speaker from Canada, and two who have completed their training in Qatar. 

We hope that this revised manuscript will meet the high standards of PLOS ONE and thank you once again for the opportunity to improve our manuscript.

Review Comments to the Author

Reviewer #1: The authors addressed all my comments and this version of the manuscript is well-written, well-presented, and support their conclusion. I recommend the acceptance of this manuscript.

Reviewer #3: The authors have addressed all my comments/suggestions. I found their responses quite satisfactory and the revised version has been much improved.

• We would like to thank the reviewers for their time and effort in reviewing our first revised manuscript. We are pleased to hear that our revisions have met their high standards.

• An adjustment was made to Table 1, the characteristics of the included studies. It was identified that the line with protocols not available did not match the PRISMA flow diagram in figure 1 (n=210). The numbers in Table 1 have been updated and accurately represent the studies that met inclusion criteria but did not have a protocol available (n=210). 

Sincerely,

Dr Daniel Rainkie, on behalf of the investigators

---

## [Editor Report · Decision Letter 2]

16 Nov 2020

Reporting and Methodological Quality of Systematic Reviews and Meta-Analysis with Protocols in Diabetes Mellitus Type II: A Systematic Review

PONE-D-20-16992R2

Dear Dr. Rainkie,

We’re pleased to inform you that your manuscript has been judged scientifically suitable for publication and will be formally accepted for publication once it meets all outstanding technical requirements.

Kind regards,

Ahmed Negida, MD

Academic Editor

PLOS ONE
---

## [Editor Report · Acceptance letter]

23 Nov 2020

PONE-D-20-16992R2 

Reporting and Methodological Quality of Systematic Reviews and Meta-Analysis with Protocols in Diabetes Mellitus Type II: A Systematic Review 

Dear Dr. Rainkie:

I'm pleased to inform you that your manuscript has been deemed suitable for publication in PLOS ONE. Congratulations! Your manuscript is now with our production department. 

Kind regards, 

on behalf of

Dr. Ahmed Negida 

Academic Editor

PLOS ONE